# Role-Perceptions of Dutch Spiritual Caregivers in Implementing Multidisciplinary Spiritual Care: A National Survey

**DOI:** 10.3390/ijerph20032154

**Published:** 2023-01-25

**Authors:** Jacqueline van Meurs, Roos Breedveld, Joep van de Geer, Carlo Leget, Wim Smeets, Robert Koorneef, Kris Vissers, Yvonne Engels, Anne Wichmann

**Affiliations:** 1Department of Spiritual and Pastoral Care, Radboud University Medical Centre, 6500 HB Nijmegen, The Netherlands; 2Department of Anaesthesiology, Pain and Palliative Medicine, Radboud University Medical Centre, 6500 HB Nijmegen, The Netherlands; 3Academic Hospice Demeter, 3731 AL De Bilt, The Netherlands; 4Agora Foundation, 3981 CK Bunnik, The Netherlands; 5Department of Care and Welfare, University of Humanistic Studies, 3512 HD Utrecht, The Netherlands; 6Professional Association of Spiritual Caregivers VGVZ, 3860 AE Nijkerk, The Netherlands

**Keywords:** spiritual wellbeing, spiritual care, multidisciplinary collaboration, communication, healthcare chaplaincy

## Abstract

Background: During the course of their disease, patients often not only suffer physical discomfort, but also face psychological, social, and spiritual challenges. However, healthcare professionals often lack the knowledge and skills to address the spiritual dimension and are in need of support for taking this responsibility. Spiritual caregivers are experts in spiritual care, but their contribution to the integration of this care by other healthcare professionals is largely unknown. Objective: The aim of this study was to investigate how Dutch spiritual caregivers view their role in increasing the integration of spiritual care in daily healthcare practice as provided by other healthcare professionals in the Netherlands, and how they address this role. Methods: An online survey was conducted from May until June 2021 among spiritual caregivers working in Dutch healthcare. Data were analysed using descriptive statistics. Results: The majority of the 174 respondents answered that they already fulfil a role in the integration of spiritual care by, for example, providing education, coaching on the job, or participating in multidisciplinary consultation. However, the majority of respondents experienced barriers to their contribution, such as confusion of terminology and use of language while collaborating with other healthcare professionals and reluctance to share information. Conclusions: While spiritual caregivers realise having a role in increasing the integration of spiritual care into daily practice of other healthcare professionals, some practices and perceptions, especially from within their own discipline, may hamper this.

## 1. Introduction

For over 40 years, the biopsychosocial model of Engel (1977) has been used as a multidimensional framework in healthcare, education and research [1]. Moreover, about two decades ago, the WHO reported an increasing awareness of the importance of faith, hope and compassion among both patients and physicians as part of the healing process and relationship [2]. In 2002 the WHO published a definition of palliative care that includes the spiritual dimension as an important dimension of full care [3]. However, in the literature, spirituality is still a highly contested term, often framed within religion and even referred to as a ‘weasel word’ [4,5]. While recognising the vagueness of the term, the number of publications on this dimension continues to increase, highlighting the importance of further developing it in health care [5,6,7,8].

In 2011, the European Association for Palliative Care (EAPC) defined a European consensus definition of spiritual care: ‘Spirituality is the dynamic dimension of human life that relates to the way persons (individuals and community) experience, express and/or seek meaning, purpose and transcendence, and the way they connect to the moment, to self, to others, to nature, to the significant and/or the sacred’ [9]. This definition takes into account the secularisation in Northern Europe, and underlines the multidimensionality of the dimension by including existential challenges (e.g., questions concerning identity, meaning and responsibility), value-based considerations and attitudes (what is most important for each person) and religious considerations and foundations (faith, beliefs and practices) [10]. Although the practice is stubborn and the discussion regarding the suitability of the term is still ongoing, this consensus definition offers room within healthcare for spirituality to be understood as broader than solely its religious aspects and the work of clergymen [9]. Since then, increasing attention to spiritual care can be found within statement documents of medical societies [11,12].

However, adequate implementation of addressing the spiritual dimension, even within the daily practice of palliative care programs, is still limited and needs more attention [13,14,15,16]. Healthcare professionals often lack the knowledge and skills to address the spiritual dimension at the right moment due to only minimal attention to this dimension in their training and education [10,15,16,17,18].

Spiritual caregivers are experts in spiritual care, while other healthcare professionals, especially when it comes to seriously ill patients, ideally provide a generalist approach to such care [19,20]. A recent literature review concluded that spiritual caregivers are actually “key personnel” when it comes to increasing the provision of spiritual care by other healthcare professionals [21]. Yet this is not today‘s reality [21,22,23]. Moreover, the integration of spiritual and medical care is generally absent in many healthcare institutions [24]. Spiritual caregivers are mostly consulted when a patient is facing an existential crisis or has a pronounced concern with finding meaning when health is challenged. Their services are often considered as only ancillary [20,21]. Relatively little outcome-oriented research has been performed on their contributions to healthcare, which is also detrimental to a more integrated approach to spiritual care [16,25].

In the Netherlands, the national multidisciplinary guideline ‘Existential and Spiritual Aspects of Palliative Care’ [26] and the palliative care quality framework [27] both state that any professional who cares for palliative patients should also pay attention to their spiritual concerns. A shared decision-making model for the elderly also explicitly recommends including what is of value to the patient [28]. In the last decade, the positive health model of Machteld Huber has become very popular to also pay attention to the patient’s contextual issues, including the spiritual dimension [29]. However, because of the earlier mentioned reasons, healthcare professionals are in need of support for taking this responsibility: in identifying what is meaningful to the patient, in addressing this in their daily care provision and in integrating such findings into decision-making, care planning and after care. Consensus on the responsibilities of spiritual caregivers in the further integration of spiritual care into healthcare is crucial [30,31]. Therefore, the aim of this study was to investigate how Dutch spiritual caregivers view their role in increasing the integration of spiritual care in daily healthcare practice as provided by other healthcare professionals in the Netherlands and how they address this role.

## 2. Materials and Methods

### 2.1. Design

We performed a survey study in Dutch. The survey was based on literature, including the Dutch national guideline Existential and Spiritual Aspects of Palliative care [26], and discussions with a team of experts, all working in an academic hospital: a professor in meaningful healthcare (YE), a postdoctoral researcher in contextual and meaningful healthcare (ABW), a medical student (RB), an associate professor ‘Innovation in Spiritual Care’ (WS), and a spiritual care provider experienced in training and educating healthcare professionals (JvM). The Checklist for Reporting of Survey Studies (CROSS) was used for reporting. See [App appA-ijerph-20-02154] for the survey.

The survey started with 12 demographic questions (e.g., age, gender, years of work experience, religion, having a mandate or endorsement by a religious or non-religious institution to undertake the work [yes/no], and if yes, which institution) (see [App appA-ijerph-20-02154]). Next, the following topics were questioned:
their view on division of tasks and collaboration with other healthcare professionals regarding providing care on the spiritual dimension (Q13, Q14, Q15, Q16, Q25; all 5-point Likert scales from disagree to agree)their actual contribution to increasing the integration of spiritual care in the care provision of other healthcare professionals (Q17, Q18; respectively a 5-point Likert scale from disagree to agree and a multiple choice question with all kinds of options)their experienced appreciation of their contribution to the integration of spiritual care in the care provision of other healthcare professionals (Q19, Q20, Q21, Q22, Q23, Q24; of which Q19, Q21, and Q23 are 5-point Likert scales from never to always; and the other open text fields)their view of working in a team with healthcare professionals (Q26, Q27, Q28; of which the first is a yes/no question and the other two multiple choices). A distinction was made between multidisciplinary and interdisciplinary collaboration. Multidisciplinary was defined as: different disciplines work together in a team but remain within their own discipline; interdisciplinary as: different disciplines work together in a team, and each team member also identifies and explores problems outside one’s own discipline. In particular, interdisciplinary collaboration requires both spiritual caregivers and other care disciplines to move into the dimension that is the primary dimension for the other.their experienced barriers in collaborating with other healthcare professionals (Q31a, Q31b, Q31c, Q32, Q33, Q34, Q35, Q36; Q31a – 31c were 5-point Likert scales from never to always; Q32 multiple choice, Q33 a 5-point Likert scale from disagree to agree, Q34 and Q35 5 point Likerts scales from never to always, and Q36 an open text field.). Some of these questions are related to language confusion. Other questions about the ‘duty of confidentiality’ that has its origins in the confessional secrecy of clergy to explore the possible differences and overlaps with confidentiality inherent to all healthcare professions.


In part of the questions, some questions were conditional; they were only asked if the participants had given a specific answer to a previous question. For example, questions regarding teamwork were only asked if the participant worked in a team. The survey took 10–15 min to complete.

### 2.2. Participants

The Dutch national Professional Association for Spiritual caregivers (“Vereniging van Geestelijk Verzorgers”, VGVZ) agreed to send the survey by e-mail to all members working in healthcare settings (hospitals, nursing and care homes, revalidation, psychiatry, care for the disabled, youth care and home care). The survey was conducted in May and June 2021. Since the VGVZ never sends email reminders about research to their members, a message on the LinkedIn of the VGVZ and through our own network was used as a reminder. All participants were a regular or prospective member of the VGVZ and working in healthcare, meaning: a hospital, nursing or care home, rehabilitation, psychiatry, youth care, with people with disabilities or in primary care.

### 2.3. Analysis

Q-ask, an online tool to anonymously send and store electronic surveys for research purposes, was used for data collection. Data were exported from Q-ask to Microsoft Office Excel 2016 and then imported to IBM SPSS Statistics 25. Participants were given a unique record-ID. All quantitative results were analysed in Dutch, using descriptive statistics. Remarks typed in the open text fields were not qualitatively analysed, but carefully read and used to illustrate quantitative figures.

Translation of the survey and respondents’ comments from Dutch to English was done by the research team. A native speaker was consulted to check some of the fragmented figures.

### 2.4. Ethics

The ethics committee MREC Oost-Nederland declared this study was not subject to the Medical Research Involving Human Subjects Act (WMO), file number 2021-8292. The information for participants was made with the guidance of the Integrated Quality System scientific research of the Radboudumc. Questions were filled out anonymously. Data will be saved for 15 years; only the study team and research management team of the department have access to it.

## 3. Results

### 3.1. Demographics

The survey was emailed to 908 spiritual caregivers, of which 414 survey emails were successfully received and opened. Two participants were excluded; one because of failure to provide informed consent and the other because the inclusion criteria were not met. A total of 174 surveys out of 414 (42%) were completed. Characteristics of study participants can be found in Table 1.

### 3.2. Respondents' View on Division of Tasks and Collaboration with Other Healthcare Professionals Regarding Providing Care on The Spiritual Dimension

Almost all respondents (167; 97%) agreed or somewhat agreed that not only spiritual caregivers but also other healthcare professionals should pay attention to spiritual care in their daily care provision. (Q13) Likewise, almost all respondents agreed or somewhat agreed (166; 97%) that identifying and exploring spiritual issues is a joint task of both the spiritual caregivers and other healthcare professionals. (Q14) One of the respondents phrased it as follows: “The person seeking care is a complete human being, who cannot and does not need to be divided into different parts depending on which healthcare professional she/he is seeing. It is up to the caregivers to identify what is going on, what they can address themselves and where they need to collaborate with or refer to colleagues from other disciplines”. If other healthcare professionals also pay attention to the spiritual dimension, this may contribute to more referrals to the spiritual counsellor. In the words of a respondent working in care for the elderly: “Since, as a spiritual caregiver, you are frequently called in by other disciplines, it is important that other disciplines also identify and explore what is going on with residents in terms of meaning”. One in four respondents (43; 26%) agreed or somewhat agreed that other healthcare professionals already sufficiently identify spiritual issues, however, two-thirds of the respondents (119; 69%) agreed or somewhat agreed that the subsequent exploration by other healthcare professionals of these issues lags behind. (Q15 & Q16) In addition, the majority of respondents (160; 93%) agreed or somewhat agreed that it is one of their tasks to contribute to the increased identification and exploration of spiritual issues by other healthcare professionals. (Q25)

### 3.3. Respondents’ Actual Contribution To Increasing The Integration Of Spiritual Care in The Care Provision of Other Healthcare Professionals

A majority of the respondents (105; 78%) agreed or somewhat agreed that they already contribute to a more integrated identification and exploration of spiritual issues by other healthcare professionals, mainly by educating their colleague healthcare professionals (105; 78%). In the words of a respondent: “In collaboration with the psychologist or social workers, they too can discuss the end of life, guilt or grief. Because I train them, I empower them. (…) I am the only spiritual caregiver and I cannot be present everywhere”. Often mentioned was the role of giving feedback to other healthcare professionals when the spiritual caregiver observed that they were missing a spiritual issue (74; 55%), participating in a multidisciplinary consultation or a consultation on a patient’s treatment plan (107; 80%) and involvement in moral case deliberations (91; 68%). (Q17) Only a minority contributes to teaching at training institutes for healthcare professionals or to coaching other healthcare professionals in the course of their work (27; 20%). Few respondents (15; 11%) coach other healthcare professionals on the job. (Q18) One respondent shared the following experience on this: “Healthcare professionals can be somewhat wary when it comes to spiritual issues, even though they often already do a lot well in this respect; that’s what I notice when I discuss something that has happened in the care they provided. It helps then to point out their instincts, intuition and gut feelings and ‘to soften their heads’. And I combine that with more information about loss, grief, life stories, hope and sources of strength. They are often open to that”.

### 3.4. Respondents’ Experienced Appreciationa of Their Contribution

Most respondents usually or always experience positive reactions when they share their expertise with other healthcare professionals (146; 84%): (Q19) “Mostly, there is recognition of the importance of spiritual care and the insight that this can be found in small things. Enthusiasm is contagious. The ability to identify what is causing friction in a given situation is also considered valuable”. But negative reactions are also sometimes (28; 16%;) or always (2; 1%) experienced: “Sometimes there is resistance, often beforehand, to spiritual care in general because of people’s own image of it, and then you can hardly get through that”. (Q21).

About one in four respondents (47; 27%) sometimes experience resistance from other healthcare professionals when they want to share their expertise: “Other healthcare professionals (such as nurses and therapists) are often busy with the primary process and experience a question about meaning as “troublesome “or “complicated”. However, most respondents (123; 72%) never or usually do not experience such resistance. “Usually other disciplines are happy with the added value I bring regarding meaning”. (Q23).

Several respondents also commented in their own words on the overlap of their work with that of psychologists or social workers. According to some of them, this can be positive but, according to others, it can also have a negative influence when sharing their expertise. For example: “What is missing here in the hospital is a team spirit among the psychosocial disciplines. There is a lot of resistance and fear that the work will be ‘‘taken away’’. I would like to see more of a team feeling; everyone has their own expertise, but there will also be some overlap. In my opinion, good communication and complementarity can only benefit healthcare”. But also: “Together with psychology and social work, we have developed a referral scheme for other healthcare professionals, so that they know where to go with what question. This is considered helpful”.

### 3.5. Respondents’ Views on Working in a Team with Healthcare Professionals

The majority of respondents (142; 83%) indicated that they work in collaboration with other healthcare professionals or form a team with them. (Q26) Of those respondents who collaborate, about half (91; 53%) said it is mainly multidisciplinary, while about a third (50; 29%) said it is interdisciplinary (Q27). Asking their opinion about the best mode of team collaboration, almost half (77; 44%) found interdisciplinary or somewhat interdisciplinary collaboration the best, while about a fourth (39; 23%) hold the view that multidisciplinary or somewhat multidisciplinary collaboration is best (Q28). One respondent argued: “Only working in a multidisciplinary way chops it up too much, recognising too little connection. Only interdisciplinary undermines the expertise too much and then the client loses the overview in care providers”. Another respondent noted: “Everyone is a specialist in their own discipline; that argues for multidisciplinary. Then there is overlap; that argues for interdisciplinary”.

### 3.6. Respondents’ Experienced Barriers in Collaborating with Other Healthcare Professionals

Language and terminology confusion regarding spiritual care was reported by two-thirds of the respondents (69; 62%) as being sometimes or usually a barrier to collaboration. (Q35) Some respondents described their own share in this: “Spiritual caregivers usually do not succeed in explaining their work in understandable language to others. They use too much of their own vocabulary”. It has also been argued that it is due to the language of the discipline, or as one respondent puts it: “Language, related to issues of meaning, is often not everyday language. Terms such as existential, spirituality, meaningfulness and ethics are wide-ranging and “vague”, and are (usually) not part of the “normal” language used by other healthcare professionals. This can widen the gap between healthcare professionals and spiritual caregivers, creating a situation where they speak to each other less often”. Not being conversant with the profession of spiritual caregivers was also mentioned: “So if people no longer think that you only come for religious matters, they think that you are half a psychologist. And then you’d better get a psychologist right away”. Only a few (5; 4%) never experienced vocabulary confusion with colleagues from other disciplines. (Q35).

Several respondents reported that their work is interpreted as that of a cleric or mainly related to religion. As one respondent put it: “There are referrers who think we are pastoral workers who come for spiritual (read: church) guidance and to talk about faith issues”.

Around a third of respondents indicated that, the ‘duty of confidentiality’, specific to the profession of a spiritual caregiver, sometimes (33; 32%) or usually (3; 3%) hinders the collaboration with other healthcare professionals (Q33), while this is not the case for half of them (usually not was answered by 53 respondents; 52%) (Q34). As one respondent put it: “Maybe it is because I have only been working as a spiritual caregiver for a short time, but I am still searching. How do I do justice to the duty of confidentiality and how can I contribute to the entire healthcare delivery process? It sometimes feels like a balancing act, where confidentiality is paramount and therefore I share little with other healthcare professionals or only in general terms”. Only a small minority (8; 13%) never experienced this as an obstacle in the collaboration with other healthcare professionals.

Accordingly, responses showed reluctance to share information about a patient with other healthcare professionals, even when this information is relevant to the medical team, and the patient has given permission to share it within the team. As one respondent put it: “I want to emphasise that I often find it a difficult balancing act. Sometimes I share with the doctor what is relevant, for example if there is a risk that a choice is made that is undesirable for the patient”. In a patient file, one in five respondents (30; 21%) never or usually does not share information about a patient with other healthcare professionals, one in five does it sometimes (30; 21%), and more than half (82; 58%) does it usually or always. (Q 31a) One of the respondents noted: “In the hospital, a short note is made in the file. In the rehabilitation centre, no information has been shared so far, but soon the spiritual caregiver will be part of an outpatient team and the outcome of the meaning assessment will be shared with the team, after the patient’s approval, via the patient file”. Verbally sixteen per cent (22; 16%) of the respondents never or usually do not share information about a patient with other healthcare professionals, and over half (85; 60%) do it sometimes, and one out of four (35; 25%) does it usually or always. (Q31b) During a multidisciplinary meeting or a meeting about the patient’s treatment plan, one in five (31; 21%) never or usually does not share relevant information about a patient, less than half of them (58; 41%) does this sometimes, and around the same percentage (53; 37%) does it usually or always. (Q31c) When they do share such information, it is often with a nurse (173; 74%) or physician (103; 73%) and half of the cases with a healthcare assistant (68; 48%) (Q32).

## 4. Discussion

Of the total 172 Dutch spiritual caregivers who completed the survey, three-quarters said they already contribute to enhancing the integration of spiritual care provided by other healthcare professionals: by teaching and coaching them on the job, by participating in multidisciplinary consultations, consultation on the patient’s treatment plan, or through participation in moral deliberation. However, it remained unknown whether these contributions are incidental or structurally embedded in their daily practice. Moreover, the survey concerned participants’ perceptions, which are subjective, and may be perceived differently by other healthcare professionals. And despite this relatively favourable picture, respondents pointed to several barriers that stand in the way of a solid profiling of their profession. These barriers were recently also identified as a research priority among Dutch spiritual caregivers [32].

Language confusion regarding spiritual care emerged as one such barrier among almost all respondents. Handzo et al. argued that such confusion occurs as spiritual caregivers use ‘their own idiosyncratic language’ [33]. It is about communication that is often cumbersome, full of jargon and gets in the way of a good exchange of information between the spiritual caregiver and other care professionals [33,34] (p. 199–204). Until a few years ago, all spiritual caregivers in healthcare had a mission from an ideological, often religious, denomination which may also have reinforced (adherence to) jargon [6]. Incidentally, Bracken et al. emphasise that all disciplines have their own use of language, so intelligibility is a key factor in interdisciplinary cooperation anyway: “Common understanding derived from shared languages, in turn, plays a vital role in enhancing the relations of trust that are necessary for effective interdisciplinary working” [35]. Adding to this, other healthcare professionals may frame the contribution of a spiritual caregiver in terminology with which the spiritual caregivers, in turn, can only partly identify [36,37]. This poses a particular challenge for spiritual caregivers, as they often see themselves as the much-needed translators of the patient’s lived experience in the world of healthcare provision [38]. Therefore, curricula for spiritual caregivers should also focus on communication with other healthcare providers. More specifically: on communication in multidisciplinary consultations and team meetings and on reporting in patient files [37]. This education should be up to date with current affairs and insights, including the ability to understand and use the practices and language of both patients and colleagues from diverse backgrounds and disciplines (e.g., nurses and physicians). This, among others, can be reached by employing language and practices that build upon commonalities rather than differences [39]. This approach can, in turn, contribute to arriving at a common, multidisciplinary language for, teamwise, identifying and responding to spiritual struggles [40].

Remarkably, the ‘duty of confidentiality’ appeared to be a barrier for about one in three respondents in sharing any information from the patient orally or via the electronic medical record, and thus in optimal collaboration with other healthcare professionals. In the Dutch context, there are actually no impediments for spiritual caregivers to be communicative in a team setting, given each healthcare professional is strongly bound by medical confidentiality outside a patient’s care team. According to the Dutch Professional Standard on Spiritual care, the spiritual caregiver is allowed to share information with other healthcare professionals if the information is relevant to and shared in the context of a joint care assignment and provided the patient has given consent [41]. This is also in agreement with the 2014 Dutch national guide ‘The Professional confidentiality in collaborations’, which was approved by eight associations of healthcare professionals [42]. Nevertheless, some spiritual caregivers experience the duty of confidentiality in daily healthcare provision as a balancing act [34] (p. 66–87). This can be attributed to the strict confidentiality of pastors, pastoral workers and clergy who visit patients from faith communities. The religious and non-religious institutions that mandate spiritual caregiving have not yet adapted this for today’s spiritual caregivers operating in healthcare and within teams. However, not sharing relevant patient information, also regarding the spiritual dimension may result in inappropriate care [34] (p. 66–87). Moreover, in communicating information obtained from a patient within the team caring for this patient, spiritual caregivers make themselves known as reliable colleagues and thus also have the opportunity to educate other healthcare professionals on the integration of spiritual care in daily practice [37]. The reported reluctance of responding spiritual caregivers to share information is incidentally in line with that of their European colleagues [37].

### Strengths and Limitations

This is the first study that examined whether guideline recommendations on interdisciplinary spiritual care are perceived to be applied in practice. Another strength of this study is that it connects perfectly with a recent survey on research priorities among Dutch spiritual caregivers, a study that highlights the importance of research into developing a stronger profile of their profession [32].

However, this study also has limitations. Part of the invitations to participate did not reach potential participants. Moreover, spiritual caregivers who are already involved in this subject matter, or have a strong opinion on it, might have been more inclined to fill in the survey and consequently have caused bias. This strengthens our findings, as we expect that non-responders will experience even more barriers. Furthermore, it remained unknown whether the reported contributions were incidental or structurally embedded in daily practice. Finally, we did not conduct a qualitative analysis of the free text comments; we only used these data illustratively. Therefore, we suggest future qualitative and quantitative, preferably international research, to get a clearer picture on this.

## 5. Conclusions

Spiritual caregivers in our survey realise that they have the potential to make important contributions to the further process of integration of spiritual care into the daily practice of other healthcare professionals. However, spiritual caregivers’ use of language or jargon and their reluctance to share information within the treatment team hamper this. We recommend further research into this, both within and outside the Netherlands. Moreover, if spiritual caregivers in the Netherlands want to avoid being regarded as an ‘allied healthcare profession’ instead of being a full member of the care team, we also recommend the Dutch professional association of spiritual caregivers and the institutions that provide spiritual caregivers the mandate to clearly define the overlap and difference of confidentiality inherent to all health care professions, and ‘clergy confidentiality’ in their policy and guidelines.

## Figures and Tables

**Table 1 ijerph-20-02154-t001:** Characteristics of respondents.

	n (%)
Number of respondents	172
Gender	
Male	53 (31)
Female	117 (68)
Other (e.g., non-binary)	2 (1)
Age	
21–30	10 (6)
31–40	18 (10)
41–50	30 (17)
51–60	67 (39)
61–70	43 (25)
71–80	2 (1)
Missing	2 (1)
Years of work experience	
0–10	98 (57)
11–20	41 (24)
21–30	24 (14)
31–40	8 (5)
Missing	1 (1)
Worldview:	
Atheism	0
Buddhism	0
Catholicism	41 (24)
Hinduism	0
Humanism	14 (8)
Islam	2 (1)
Judaism	0
Protestantism	67 (39)
A combination of beliefs	39 (23)
None	1 (1)
Other	8 (5)
Endorsement or mandate	
by a religious or worldview institution	90 (52)
from the Council Non-Denominational Spiritual Caregivers (RING-GV) ^1^	57 (33)
Neither	25 (15)

^1^ This council screens the competence of all spiritual caregivers who do not have a mission from a church or worldview institution.

## Data Availability

The anonymised dataset is available from the corresponding author upon reasonable request.

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
