# Peer review of "Role-Perceptions of Dutch Spiritual Caregivers in Implementing Multidisciplinary Spiritual Care: A National Survey"

_ijerph, 2023, doi:10.3390/ijerph20032154_

Round 1
Reviewer 1 Report
Thank you for a well written and interesting piece of work which will have relevance beyond the context studied. I think the article is publishable as it stands. It is well referenced, coherent and significant for the field.
Author Response
We are delighted with this very positive reply, and would like to thank the reviewer for the effort put into reviewing our manuscript.

Reviewer 2 Report
This is a timely and very well written paper. It was a pleasure to read and review, and I recommend that it should be accepted with some relatively minor revisions as set out below.
The paper reports on a study that set out to investigate how Dutch spiritual caregivers view their role in increasing the integration of spiritual care in daily healthcare practice as provided by other healthcare professionals in the Netherlands, and how they address this role. Spirituality in the health-care sector broadly is an area in which, until more recently, there has been relatively little research, and this is submission makes and important and appropriate contribution to the field.
The authors draw on the definition of spirituality as set out by the European Association for Palliative Care (EAPC). This definition states that, ‘Spirituality is the dynamic dimension of human life that relates to the way persons (individual and community) experience, express and/or seek meaning, purpose and transcendence, and the way they connect to the moment, to self, to others, to nature, to the significant and/or the sacred.' . While this definition is fine, I wonder whether there
needs to be a short discussion, drawing on the contemporary academic literature, to describe spirituality in a little more detail, and showing how the EAPC has subsequently arrived at its definition. Spirituality is a contested term, and much of the literature in this field suggests that it is a term that is better described since it evades a precise definition. The EAPC definition could then be a means by which to ‘bring together’ the key points made in some of the contemporary academic literature. I stress, this inclusion would not need to be lengthy. But it would enhance the quality of
the submission.
An online survey was conducted among spiritual caregivers working in 34 healthcare settings, and data were analysed using descriptive statistics. The findings are succinctly set out and appropriately discussed, indicating that three-quarters of respondents said they already contribute to enhancing the integration of spiritual care provided by other healthcare professionals: by teaching and coaching them on the job, by participating in multidisciplinary consultations, consultation on the patient's treatment plan, or through participation in moral deliberation. However, it remains
unknown whether these contributions are incidental or structurally embedded in their daily practice.
In spite of this relatively favourable situation, respondents pointed to several barriers that stand in the way of a solid profiling of their profession.
In light of these findings, what might the authors recommend for future practice? What might the authors recommend as a means by which to ascertain whether the contributions of spiritual caregivers are incidental or structurally embedded in their daily practice? How might some of the barriers standing in the way of profiling spiritual caregivers’ profession be addressed? And are there any recommendations for further research that stem from this project? I think this is an important section that is currently missing from is otherwise a fine piece of writing.
The strengths and limitations of the study are appropriately acknowledged. This is just a suggestion – and other reviewers may not find this a problem – but I wonder whether
the more traditional ‘in-text’ referencing style (e.g., APA, or Harvard) might render the paper more fluent and accessible for the reader. This would entail (if the suggestion is adopted) placing each of the entries in the reference list in alphabetical order, instead of the actual order in which they are referenced in the text itself.
Thank you for this well written and timely submission!
Author Response
This is a timely and very well written paper. It was a pleasure to read and review, and I recommend that it should be accepted with some relatively minor revisions as set out below. (…) Spirituality in the health-care sector broadly is an area in which, until more recently, there has been relatively little research, and this submission makes and important and appropriate contribution to the field.
We are happy you find our paper important and a well-timed contribution to the field, and think it should be accepted after minor revisions. Below, we point-by-point respond to your suggestions.
- The authors draw on the definition of spirituality as set out by the European Association for Palliative Care (EAPC). This definition states that “Spirituality is the dynamic dimension of human life that relates to the way persons (individual and community) experience, express and/or seek meaning, purpose and transcendence, and the way they connect to the moment, to self, to others, to nature, to the significant and/or the sacred.” While this definition is fine, I wonder whether there needs to be a short discussion, drawing on the contemporary academic literature, to describe spirituality in a little more detail, and showing how the EAPC has subsequently arrived at its definition. Spirituality is a contested term, and much of the literature in this field suggests that it is a term that is better described since it evades a precise definition. The EAPC definition could then be a means by which to ‘bring together’ the key points made in some of the contemporary academic literature. I stress, this inclusion would not need to be lengthy. But it would enhance the quality of the submission.
We cannot agree more with the reviewer, and think this is a very relevant point. Therefore, the texts below are integrated in or added to the manuscript text about the EAPC definition:
“However, in the literature spirituality is still a highly contested term, often framed within religion and even referred to as a ‘weasel word’ [1, 2]. At the same time, while recognising the vagueness of the term, the number of publications on this dimension increases, highlighting the importance of further developing it in health care [2-5].” - Page 2, lines 53-57
“This definition takes into account the secularisation in Northern Europe, and underlines the multidimensionality of the dimension by stressing the fact it consists of existential challenges (e.g. questions concerning identity, meaning and responsibility), value based considerations and attitudes (what is most important for each person) and religious considerations and foundations (faith, beliefs and practices) [6]. Although practice is stubborn and the discussion regarding the suitability of the term is still ongoing, this consensus definition offers room within healthcare for spirituality not to be narrowed down to its religious aspects and the work of clergymen [7]. Since then, increasing attention to spiritual care can be found within statement documents of medical societies [8] [9].” - Page 2, lines 63-68
- An online survey was conducted among spiritual caregivers working in 34 healthcare settings, and data were analysed using descriptive statistics. The findings are succinctly set out and appropriately discussed, indicating that three-quarters of respondents said they already contribute to enhancing the integration of spiritual care provided by other healthcare professionals: by teaching and coaching them on the job, by participating in multidisciplinary consultations, consultation on the patient's treatment plan, or through participation in moral deliberation. However, it remains unknown whether these contributions are incidental or structurally embedded in their daily practice.
In spite of this relatively favourable situation, respondents pointed to several barriers that stand in the way of a solid profiling of their profession. In light of these findings,
- What might the authors recommend for future practice?
Indeed, based on our study findings we conclude that although the majority of respondents reported they contribute to enhancing the integration of spiritual care multi- or interdisciplinary, still barriers are experienced in doing this. These barriers mainly related to (1) language confusion and (2) duty of confidentiality/reluctance to share information.
Regarding the first, we recommend further research uncovering how to deal with this issue in practice (page line 9, line 391). Also we recommend curricula for spiritual caregivers focus on communication more – please find our response under c. In the Netherlands, a study – initiated by the Dutch professional association of spiritual caregivers – about the use of language concerning this dimension in Dutch education will start this year. Main goal of this project: to align language use over educational institutions.
Regarding the second, we recommend the Dutch professional association of spiritual caregivers and the sending agencies, to clearly define the overlap and difference of confidentiality, inherent to all health care professions, and ‘clergy confidentiality’ in their policy and guidelines (see page 9, lines 393-396).
b. What might the authors recommend as a means by which to ascertain whether the contributions of spiritual caregivers are incidental or structurally embedded in their daily practice?
Indeed, our study did not include whether or not the self-reported contributions of spiritual caregivers were structurally embedded in daily practice. We agree this would be valuable information to gather in future research – for example by also asking about both time investment and the specific focus of the contributions (education, coaching, consultation, etc.). Therefore, we added the below text to the limitations section:
“Furthermore, it remained unknown whether the reported contributions were incidental or structurally embedded in daily practice. Therefore, we suggest future qualitative and quantitative, preferably international research to get a clearer picture on this.” - Page 9, lines 382-385
c. How might some of the barriers standing in the way of profiling spiritual caregivers’ profession be addressed?
We agree this is a very relevant point – as we assume this will influence time allocation in daily practice. Therefore, in the conclusion section it is recommended professional association and sending agencies and (as our study concerns the Netherlands) the RING (Dutch abbreviation for Council for Institutionally Non-Aligned Spiritual Caregivers) to come to the table and provide frameworks regarding these issues.
“Moreover, if spiritual caregivers in the Netherlands want to avoid being regarded as an ‘allied healthcare profession’, we also recommend the Dutch professional association of spiritual caregivers and the sending agencies, to clearly define the overlap and difference of confidentiality, inherent to all health care professions, and ‘clergy confidentiality’ in their policy and guidelines.” - Page 9, lines 392-396
Moreover, we also think it is relevant to, in educational curricula for spiritual caregivers, focus on communication with other healthcare providers. More specifically: on communication in multidisciplinary consultations and team meetings, and on reporting in patient files. This relevance also emerges from recent literature (see reference 10). Therefore, we added the below text to the manuscript:
“Therefore, curricula for spiritual caregivers should also focus on communication with other health care providers. More specifically: on communication in multidisciplinary consultations and team meetings, and on reporting in patient files [10]. This education should be up to date with current affairs and insights, including the ability to understand and use the practices and language of both patients and colleagues from diverse backgrounds and disciplines (e.g. nurses and physicians). This among others can be reached by employing language and practices that build upon commonalities rather than differences [11]. This approach can in turn contribute to arriving at a common, multidisciplinary language for, teamwise, identifying and responding to spiritual struggles [12].” - Page 8, lines 339-347
d. And are there any recommendations for further research that stem from this project?
Next to the above described recommendations, we think it would be valuable to conduct similar research internationally, so that ways of working, experiences and proceedings can be compared on a larger scale. We now integrated the below text in the manuscript.
“Furthermore, it remained unknown whether the reported contributions were incidental or structurally embedded in daily practice. Therefore, we suggest future qualitative and quantitative, preferably international research to get a clearer picture on this.” - Page 9, lines 382-385
I think this is an important section that is currently missing from is otherwise a fine piece of writing.
- The strengths and limitations of the study are appropriately acknowledged. This is just a suggestion – and other reviewers may not find this a problem – but I wonder whether the more traditional ‘in-text’ referencing style (e.g. APA or Harvard) might render the paper more fluent and accessible for the reader. This would entail (if the suggestion is adopted) placing each of the entries in the reference list in alphabetical order, instead of the actual order in which they are referenced in the text itself.
This indeed might be easier for readers. However, in the reference style adopted while drafting the paper, we followed the referencing instructions of the IJERPH. These instructions state that references must be numbered in square brackets [ ]. For embedded citations in the text with pagination, both parentheses and brackets have to be used to indicate the reference number and page numbers; for example [5] (p. 10). or [6] (pp. 101–105).
Thank you for this well written and timely submission!

Reviewer 3 Report
My only suggestion is to elaborate in the conclusion the need for education about interreligious, research-literate spiritual care. The authors could explore whether this recent publication (especially chapters on meaning-making and interpersonal competencies) could be relevant in educating spiritual caregivers in the Netherlands:
Rambo, S., & Cadge, W. (Eds.). (2022). Chaplaincy and spiritual care in the twenty-first century: An introduction. University of North Carolina Press.
A research-literate approach could find a common language for identifying and responding to spiritual struggles. Research on such struggles is summarized in this publication:
Pargament, K., & Exline, J. J. (2022). Working with spiritual struggles in psychotherapy: From research to practice. Guilford.
Author Response
Thank you for these valuable additions. We agree education is of the utmost importance in battling current barriers and changing the future of clinical practice, and therefore added the text below to the manuscript.
“Therefore, curricula for spiritual caregivers should also focus on communication with other health care providers. More specifically: on communication in multidisciplinary consultations and team meetings, and on reporting in patient files [10]. This education should be up to date with current affairs and insights, including the ability to understand and use the practices and language of both patients and colleagues from diverse backgrounds and disciplines (e.g. nurses and physicians). This among others can be reached by employing language and practices that build upon commonalities rather than differences [11]. This approach can in turn contribute to arriving at a common, multidisciplinary language for, teamwise, identifying and responding to spiritual struggles [12].” - Page 8, lines 339-347
